# Identification of Heat-Tolerant Genes in Non-Reference Sequences in Rice by Integrating Pan-Genome, Transcriptomics, and QTLs

**DOI:** 10.3390/genes13081353

**Published:** 2022-07-28

**Authors:** Samuel Tareke Woldegiorgis, Ti Wu, Linghui Gao, Yunxia Huang, Yingjie Zheng, Fuxiang Qiu, Shichang Xu, Huan Tao, Andrew Harrison, Wei Liu, Huaqin He

**Affiliations:** 1College of Life Sciences, Fujian Agriculture and Forestry University, Fuzhou 350002, China; samuel_tareke@163.com (S.T.W.); 3200537077@fafu.edu.cn (T.W.); 1190514014@fafu.edu.cn (L.G.); 1190514025@fafu.edu.cn (Y.H.); 1200525042@fafu.edu.cn (Y.Z.); 1200514046@fafu.edu.cn (F.Q.); xushichang@fafu.edu.cn (S.X.); taohuan@fafu.edu.cn (H.T.); weilau@fafu.edu.cn (W.L.); 2Department of Mathematical Sciences, University of Essex, Colchester, CO4 3SQ, UK; harry@essex.ac.uk

**Keywords:** rice (*Oryza sativa* L.), heat stress, pan-genome, single nucleotide polymorphisms (SNPs), presence/absence variation (PAV)

## Abstract

The availability of large-scale genomic data resources makes it very convenient to mine and analyze genes that are related to important agricultural traits in rice. Pan-genomes have been constructed to provide insight into the genome diversity and functionality of different plants, which can be used in genome-assisted crop improvement. Thus, a pan-genome comprising all genetic elements is crucial for comprehensive variation study among the heat-resistant and -susceptible rice varieties. In this study, a rice pan-genome was firstly constructed by using 45 heat-tolerant and 15 heat-sensitive rice varieties. A total of 38,998 pan-genome genes were identified, including 37,859 genes in the reference and 1141 in the non-reference contigs. Genomic variation analysis demonstrated that a total of 76,435 SNPs were detected and identified as the heat-tolerance-related SNPs, which were specifically present in the highly heat-resistant rice cultivars and located in the genic regions or within 2 kbp upstream and downstream of the genes. Meanwhile, 3214 upregulated and 2212 downregulated genes with heat stress tolerance-related SNPs were detected in one or multiple RNA-seq datasets of rice under heat stress, among which 24 were located in the non-reference contigs of the rice pan-genome. We then mapped the DEGs with heat stress tolerance-related SNPs to the heat stress-resistant QTL regions. A total of 1677 DEGs, including 990 upregulated and 687 downregulated genes, were mapped to the 46 heat stress-resistant QTL regions, in which 2 upregulated genes with heat stress tolerance-related SNPs were identified in the non-reference sequences. This pan-genome resource is an important step towards the effective and efficient genetic improvement of heat stress resistance in rice to help meet the rapidly growing needs for improved rice productivity under different environmental stresses. These findings provide further insight into the functional validation of a number of non-reference genes and, especially, the two genes identified in the heat stress-resistant QTLs in rice.

## 1. Introduction

The growth of the world population needs our best efforts to increase crop production by 100% before 2050 [1]. However, a number of environmental factors, such as light, water, temperature, etc., significantly affect the production of crops. Due to global climate change, high temperatures, in particular, have become one of the major disasters affecting crop production and quality [2]. Rice is one of the most widely produced crops and is consumed as a staple food by a large part of the world’s human population, providing more than 20% of calories (FAO 2016 statistics). It has been cultivated in a wide range of climatic environments. The majority of the world’s top rice producers are mainly located in the tropics and subtropics, where the temperature is high during the rice crop season. High-temperature stress is a complex interaction between temperature intensity, duration, rapidity, and plant growth stage. Damage from extreme high temperatures is particularly severe when it occurs during the crop’s critical developmental stages, particularly the reproductive period. The optimal temperature for rice plants during the reproductive stage is 20–30 °C, but temperatures surpassing 35 °C have critical negative effects on rice growth. High daytime temperatures in some of the major tropical rice-growing regions are already close to the threshold, beyond which yield begins to decline [3]. One of the fundamental measures to overcome the yield loss of rice under high-temperature stress is to breed heat-tolerant rice varieties [4].

The heat tolerance of plants refers to the ability of plants to avoid and endure high-temperature adversity. The tolerance of high temperatures in rice germplasm resources has been identified in both *Indica* and *Japonica* subspecies [5,6]. The *Japonica* rice cultivars, Akitakomachi, Nipponbare, Hitomebore, and Todorokiwase, are classified as heat-tolerant genotypes [6,7,8] while the *Indica* cultivars, IR24, IR36, Ciherang, ADT36, BG90-2, Dular, Huanghuazhan, AUS17, M9962, Sonalee, Carreon, Dular, N22, OS4, P1215936, HT54, Sintiane Diofor, and AUS16, are known as heat-tolerant genotypes [7,8,9,10,11]. In many research works, N22 has been used as an excellent heat-tolerant rice variety [3,12,13]. Giza178, an Egyptian cultivar developed from *Japonica*–*Indica* cross breeding, has also shown considerable heat tolerance during the booting stage and the flowering stage [8]. Accurate evaluation of the thermotolerant degree of these rice cultivars and successful transfer of these thermotolerant traits into specific cultivars with good agronomic performance is of great importance for rice producers.

Apart from the thermotolerance phenotypic studies, genetic studies have been conducted to dissect and understand the mechanisms of heat stress resistance and discover heat-resistant genes or quantitative trait loci (QTLs) and apply them to thermotolerance breeding. Multiple genetic studies have shown that the heat tolerance of rice is a multigenic trait that varies with the development stages and plant tissues [14,15]. With the advance of molecular marker technology, the detection of heat-tolerant QTLs and investigation of its genetic effects has become possible. Multiple heat-responsive QTL-related traits, such as spikelet sterility, yield, flowering time, pollen fertility, and stay green, were mapped on all 12 rice chromosomes [13,16,17,18,19,20,21,22,23,24,25,26,27,28,29,30]. With the application of single nucleotide polymorphisms (SNPs) in the third generation of molecular markers, genome-wide association study (GWAS) has emerged as a tool to resolve complex trait variation down to the sequence level by exploiting historical and evolutionary recombination events at the population level [31]. This approach has been successfully applied to dissect the number of important agronomic traits in plants. Anuj et al. examined 190 rice accessions, including *Indica* and *Japonica* sub-species, and identified 966 new heat stress-resistant loci linked with the panicle length and number of spikelets [32]. Lafarge et al. conducted GWAS to detect 14 loci associated with heat stress responses [33]. Similarly, Kilasi et al. also found multiple QTLs for different traits under heat stress with varying phenotypic contributions [34].

Access to plant genomes has revolutionized the opportunities to discover specific genes and their subsequent associated traits. The efforts to dissect the genetic architecture of agronomically important traits in rice, such as QTL, GWAS, and genomic prediction, have been carried out primarily at the level of SNPs [35,36]. These SNP discovery methods were solely based on a single reference genome, which cannot cover the entire gene content of a species due to structural variations, such as gene presence/absence variations (PAVs) or copy number variations (CNVs) [37]. To address this issue, pan-genomes have been constructed to detect the PAVs in a number of plants, including maize, soybean, rice, tomato, wheat, sorghum, pigeon peas, and *Brassica* [38,39,40,41,42,43]. Multiple studies have also uncovered that these PAVs are associated with environmental adaptation of plants, such as abiotic and biotic stress tolerances [44,45,46].

Thus, a pan-genome comprising all genetic elements is crucial for comprehensive variation study among the heat stress-resistant and -susceptible rice varieties. Therefore, in this paper, we firstly constructed a rice pan-genome reference from 60 heat-responsive rice cultivars using a pan-genome iterative mapping and assembly approach. Secondly, we detected the presence and absence of variation (PAV) and SNPs in the tested rice accessions. Thirdly, we identified the SNPs specific to the highly resistant rice cultivars and compared the results with the outcome of the comparative transcriptome analysis we performed on multiple RNA-seq datasets to detect the potential candidate heat stress tolerance genes in the non-reference sequences. Finally, the heat stress tolerance genes were also identified by mapping the pan-genes to known heat stress-tolerant QTLs in multiple rice reference genomes. 

## 2. Materials and Methods

### 2.1. Data Collection

The genome resequencing data of 60 rice varieties with different heat stress tolerance was downloaded from the 3000 rice genomes project (3K RGP) and other heat stress-related studies. The heat stress response of each variety was identified from previously conducted studies and the heat stress response of each variety was collected from each respective study (Appendix A). Based on the respective study conducted, the response of each rice cultivar was evaluated according to its panicle development and spikelet formation recorded with heat stress exposure. Accordingly, the rice cultivars were classified as highly tolerant (with spikelet fertility >65%), tolerant (with spikelet fertility 50% to 65%), moderately tolerant (35% to 50%), susceptible (15% to 35%), and highly susceptible (≤15%).

### 2.2. Pan-Genome Assembly and Annotation

By using the iterative mapping and assembly approach [40], the pan-genome reference was constructed from the genome resequencing data of 60 rice varieties, including 45 heat-resistant and 15 heat-susceptible rice varieties. There were 2 *Admixture*, 6 *Indica*, and 7 *Japonica* in the heat-susceptible group while the heat-resistant group was composed of 1 *Aus*, 2 *Admixture*, 11 *Japonica*, and 31 *Indica* rice varieties.

The pan-genome was constructed by mapping the sequence reads individually to the Nipponbare reference genome by using Bowtie2 v2.4.2 with (-I 0 -X 1000) options [47], and the unmapped reads were assembled using MaSuRCA v3.4.2 [48] to produce additional reference sequences. Then, the assembled sequences were rechecked for any redundancy with the reference genome sequence using BLAST v2.10.0. The assembled contigs sequences were then compared to the National Centre for Biotechnology Information (NCBI) nt database using BLAST v2.10.0 to filter out non-green plant sequences. Subsequently, contigs with the best hit to non-green plants sequences were removed. Additionally, redundant sequences were removed using the CDHIT tool. The remaining newly assembled contigs >500 b in length were annotated using MAKER2 [49]. The assembled sequences were annotated by combining evidence-based ab initio gene prediction with the SNAP [50] and Augustus [51] tools. Publicly available assembled rice ESTs (284,186) from (www.plantgdb.org (accessed on 21 February 2021)) and 4 rice RNA-seq data sets (PRJNA79825, PRJDA67119, PRJNA508820, and PRJNA562794) and plant proteins (43,287) from NCBI were used as evidence. Finally, functional annotations of the predicted genes were performed using the Blast2GO tool [52] and the eggnog [53] functional annotation tools. Gene ontology (GO) terms were assigned according to the GO terms of the best hit of each gene. 

### 2.3. Gene Presence/Absence Variation and Pan-Genome Modeling

We performed gene presence and absence analysis on the 56 rice cultivars with a read depth of greater than 10×. We first aligned the raw reads of these 56 rice varieties to the pan-genome sequence using bowtie2 v2.4.2 with (-I 0 -X 1000) options [47]. Then, the gene PAV profile was calculated using the SGSgeneloss package with the criteria of at least 5 covered reads and a lost cutoff of 20% (minCov = 5 and lostCutoff = 0.2) [54]. A gene was considered as present if >80% of the gene body was covered by at least 5 reads; otherwise, it was considered as absent. To model the pan-genome gene growth, the mean count for each sample size of core and pan-genome genes present in all possible combinations of 56 accessions was plotted. The pan-genome genes’ and core genes’ expansion was modeled using the PanGP modeling tool [55]. To investigate the relationship between the heat stress responsive cultivars based on the PAV, the Jaccard similarity index was calculated and a tree was constructed using an in-house Python script.

### 2.4. Linking the Known Heat-Resistant QTLs with the Predicted Genes

Previously conducted QTL studies were used to map the pan-genome genes in known heat stress tolerance QTLs. Consequently, 46 known heat stress tolerance QTLs in rice were collected (Appendix A). The sequences of QTL markers and primer pairs were downloaded from the Gramene QTL database (https://archive.gramene.org/qtl/ (accessed on 2 March 2021)). BLAST was used to map the marker positions on the twelve cultivated Asian rice reference genomes (Appendix A) [56]. Finally, the genes in the rice pan-genome were mapped to the QTL regions in each reference genome, and homologous heat stress-tolerant genes were identified. 

### 2.5. Processing of RNA-seq Datasets

The RNA-seq datasets were downloaded from the SRA archive of the NCBI database. First, the fastq-dump tool available in the SRA-Toolkit version 2.8.2 (http://ncbi.github.io/sra-tools (accessed on 26 October 2020)) was run with the options “–gzip” and “–split-spot” to split the fastq reads. Residual adaptor sequences at both 5′ and 3′ ends were removed from the raw reads using the default parameters of the Fastp trimming and cleaning tool [57]. To deliver accurate quantitative transcript-specific expression data from the RNA-seq datasets, we used STAR aligner [57] to align to the pan-genome and count the transcript information with (–outFilterMismatchNmax 999 –alignIntronMin 20 –alignIntronMax 10,000 –quantMode GeneCounts –alignMatesGapMax 1,000,000) options. Finally, differential expression analysis was performed using DESeq2 and significant differentially expressed genes were defined as those with a false discovery rate (padj) < 0.05 [58].

### 2.6. SNP Discovery and Annotation

Variants were identified based on the GATK best practices for SNP/Indel discovery [59]. GATK version 4.2.1.0 was employed for all steps. Initially we performed the data preprocessing for the variant discovery. Firstly, a quality check of the resequencing data of all the rice varieties was conducted and the low-quality reads were trimmed using the Fastp trimming tool [60]. Secondly, whole-genome sequence reads were mapped to the pan-genome using Bowtie2 v2.4.2 [47]. The resulting SAM files were then converted to BAM format using samtools [61], followed by the removal of duplicate reads using picard tools v2.30 [62]. Then the data preprocessing was completed by recalibrating the reads using the GATK BaseRecalibrator and ApplyBQSR tools and making it ready for variant calling. Variants were then called on a per sample basis using GATK HaplotypeCaller, and variants were then consolidated in a joint calling step with GenotypeGVCFs. Variants of low quality were then filtered out using the GATK VariantFiltration tool with the default criteria for filtering SNPs and indels. Subsequently, the variants missing in at least 80% of the varieties and MAF of less than 0.05 were filtered out using the vcftools tool [63]. All variants were annotated for their potential effects using SnpEff 4.3 t with the annotation database built from the pan-genome gene set [64]. Finally, we selected all the SNPs that were specific to the heat stress-resistant cultivars. An SNP-based phylogenetic tree was then constructed by using the SNPrelate and ape R-packages and, finally, the tree was plotted using the ITOL (https://itol.embl.de/ (accessed on 29 May 2022)) online phylogenetic plotting tool [65]. 

## 3. Results

### 3.1. Pan-Genome Assembly and Annotation

In this research work, we gathered 60 rice accessions with different tolerance of heat stress. Based on the criteria mentioned in the data collection section of the Material and Methods, the heat stress response recorded during panicle development and spikelet fertility, the rice cultivars were classified as 3 highly resistant, 36 resistant, 6 moderately resistant, 10 susceptible, and 5 highly susceptible to heat stress. The heat-responsive rank and genome resequencing depth are provided in the Appendix A. 

The genome resequencing data of these 60 rice varieties was used to build the rice pan-genome. After mapping the short read sequences to the Nipponbare genome, a total of 525 Mb non-reference sequences were obtained. The removal of contaminants (non-green plant sequences) and redundant contigs resulted in 38,189 non-reference contigs with a total length of 71,740,214 bp. In the final assembled non-reference sequences, using ab initio gene prediction tools and additional RNA-seq data, protein sequences, and EST sequences, a total of 1141 fully annotated genes were predicted (Appendix A). 

### 3.2. Core and Variable Genes in the Pan-Genome

The PAV analysis was conducted on the whole-genome resequencing reads of 56 rice accessions with a sequencing depth greater than 10x. Subsequently, the presence and absence profile of each gene was calculated using the SGSgeneloss package [54]. The majority of genes were core genes, 31,046 (79.61%) of the pan-genome gene set, which were shared in all the accessions. In total, 7952 (20.39%) of the pan-genome gene set were identified as variable genes, which were absent in at least one individual rice accession (Appendix A). The size of the pan-genome expanded with each additional line to 38,998 genes while the number of core genes decreased, and variable genes increased with each added accession to 31,046 (Figure 1A). A total of 26 genes were present in a single rice variety while the remaining variable genes were observed in more than one variety. The comparison of the gene’s presence in the heat-resistant and -susceptible rice cultivars showed that 53 genes were uniquely present in the heat-resistant rice cultivars, including 5 genes from the reference contigs whereas 48 were from the additional non-reference contigs. Additionally, the comparison of the gene length between the variable genes and core genes showed that core genes were longer than the variable genes and a relatively higher number of exons were observed in the core genes (Figure 1B,C). The sequences of the additional annotated genes were generally shorter than the genes from the reference sequences, with an average length of 1.94 Kbp, where the number of exons per gene varied between 1 and 9, with an average length of exons of 350 bp.

The PAV-based relationships of the rice cultivars were accessed using the Jaccard similarity index. The Jaccard similarity index of the genes’ presence/absence variation varied between 0.9 and 1, suggesting that there was a close relationship between the different accessions (Figure 2). However, the tree constructed from this similarity index showed that the rice cultivars TN1 and BG90-2 were separated from the other accessions in one cluster, which mainly resulted from the lowest number of genes being present in these two accessions, with 34,999 and 35,003, respectively. On the other hand, the remaining accessions clustered into two distinct clusters. The first cluster contained 18 rice accessions and the second cluster contained the remaining 36 accessions. The majority of the rice accessions in the first cluster were *Japonica*, containing 8 susceptible and 10 tolerant accessions. In total, 38,601 pan-genome genes were shared in this clade while 159 genes were present only in the resistant accessions. On the other hand, in the second cluster, the majority of the rice accessions belonged to *Indica*, including 29 tolerant and 7 susceptible accessions. Similarly, 38,860 genes were shared in these accessions and 370 genes were unique to the resistant accessions. The number of heat-resistant accessions sharing these unique genes varied from 1 to 14.

### 3.3. Functional Annotation of Genes

GO-based enrichment analysis of the predicted genes showed that 491 of them were involved in biological processes, among which the significantly enriched category included 38.9% involved in the response to stress, 20% in involved in biological regulation, and 11.8% in signal transduction functions. In the molecular function category, 535 genes were identified in which the significant category included 59% associated with binding and 45.7% associated with catalytic activity. In total, 485 of the predicted genes were also annotated to be involved in the cellular component functions, in which the genes associated with the organelle part (53%), membrane part (10%), and protein-containing complex (8%) were among the significantly enriched gene categories (Figure 3).

### 3.4. SNP Analysis

We identified a large number of variants (SNPs) by mapping the heat stress-responsive cultivars’ whole-genome sequence reads to the pan-genome reference using the GATK tools. After filtering out the low-quality SNPs, SNPs with MAF ≤0.05, and SNPs with missing genotypes over 80%, a total of 5,059,798 biallelic SNPs were detected, among which 191,187 SNPs were identified in the non-reference contigs. It was observed that chromosome 1 had the highest number of SNPs (543,803), followed by chromosomes 4, 8, and 11. On the other hand, chromosome 9 contained the fewest number of SNPs (310,316). The SNP density comparison revealed that chromosome 8 had the highest number of SNPs per Kbp with 26.27/Kbp, followed by chromosome 10, 11, and 12, and chromosome 3 had the lowest density of SNPs per Kbp with 17.12/Kbp. The SNP density in the non-reference contigs (8.72/Kbp) was less than that in the reference genome (21.69/Kbp). The SNPs in each variety varied from 98,704 in HINUKARI to 2,211,361 in IR36 (Figure 4). N22 had the highest number of SNPs with 85,186 in the non-reference contigs, followed by VANDANA with 80,799 and DULAR with 79,322.

To understand the relationship of the tested 60 rice varieties, a neighbor joining (NJ) tree was constructed using the SNPs. The accessions were categorized into the rice sub-species *Indica*, *Japonica*, *Aus*, and *Admix*. However, the phylogenetic relationships based on the SNPs grouped the varieties into two major clusters, which are in agreement with their population classification. The first cluster contained 19 rice varieties, in which the majority belonged to *Japonica* species, whereas the second cluster included the remaining 41 rice varieties, with the majority belonging to *Indica* species (Figure 4). To further understand the variation within each cluster, we analyzed the SNPs separately in each cluster. The cluster-based SNP classification showed that 2,373,085 SNPs were detected in cluster-1 and 4,763,997 SNPs were detected in cluster-2 (Table 1). It was also observed that 2,686,715 SNPs were specific to cluster-2 and 295,803 SNPs specific to cluster-1 whereas 2,077,283 SNPs were shared between the two clusters (Table 1).

The classification of the above 5,059,798 SNPs illustrates that the highest number of SNPs were located in the intergenic regions (45%), followed by the upstream (29.9%), downstream (14.4%), exonic (3.92%), and intronic (3.83%) regions (Table 1). Missense SNPs, which could change the coding amino acid sequence, accounted for only 2.84%, and the fraction of low-effect variants was 2.5%. Among the SNPs located in the coding sequences, 50.64% were nonsynonymous and 49.36% were synonymous. Meanwhile, the large-effect SNPs, which could modify splice sites and stop or start codons, represented the smallest class, with only 1936 (0.038%).

### 3.5. Identification of Heat Stress Tolerance-Related SNPs and Genes

To further identify the heat tolerance-related variants and candidate genes, we investigated the genes that harbored the SNPs specific to the heat-resistant rice cultivars. Subsequently, we placed more emphasis on the high-impact SNPs, SNPs in the genic regions, and SNPs located within 2 kbp upstream and downstream of the genes. Using these criteria, we identified a total of 146,773 SNPs, including 2427 SNPs in 435 non-reference genes. Additionally, we performed further variant filtering to screen the SNPs specific to the highly heat-resistant cultivars. As a result, 76,435 SNPs were identified to be specific to the highly resistant cultivars, including 827 SNPs in 187 non-reference genes, which were named heat tolerance-related SNPs. The 76,435 SNPs were annotated as 162 high-impact SNPs (splice site acceptor, splice site donor, start lost and stop gained), 5046 moderate-impact SNPs (non-synonymous), 66,575 modifier SNPs, and 4458 low-impact SNPs (Table 2). 

### 3.6. Meta-Analysis of Comparative Transcriptomic Data

Four RNA-seq datasets of rice under heat stress, PRJNA633211, PRJNA610667, PRJNA604026, and PRJNA508820, were downloaded from the NCBI database. Each dataset contained the transcriptome data of two rice cultivars with a contrasting response to heat stress and each sample had a minimum of two replicas (Table 3). After trimming the raw reads of each dataset, clean data were obtained with a quality of 91.6% reads over Q30. All the clean reads were then mapped to the pan-genome gene set and the alignment and mapping of genes were greater than 91.2% and 79.0%, respectively. Finally, differential expressing genes (DEGs) in each sample were examined using the threshold of the false discovery rate (padj ≤ 0.05) and log2foldchange (|LOG2FC| ≥ 1). After comparing the heat-tolerant cultivar with the heat-susceptible cultivar in each experiment, we obtained a total of 21,706 DEGs in PRJNA604026, 10,599 DEGs in PRJNA633211, 7624 DEGs in PRJNA610667, and 5100 DEGs in the PRJNA508820 dataset (Table 3). As shown in Table 3, the number of DEGs varied among the different experiments, which might be due to the different varieties, experimental design, and technology used.

To further investigate the genes associated with heat stress tolerance, we screened the DEGs with heat tolerance-related SNPs in all the datasets and excluded the DEGs with a contradicting expression profile among the different studies. Consequently, we were able to identify 3214 upregulated and 2212 downregulated genes with heat tolerance-related SNPs in one or more RNA-seq datasets (Figure 5). Among these, 24 DEGs were located in the non-reference contigs, including 15 upregulated and 9 downregulated genes (Figure 6). 

Based on the functional and GO-based annotation, we found that some of the 24 non-reference genes with heat stress tolerance-related SNPs were homologous to the genes in wild rice. Calmodulin-binding protein 60 A-like *(maker_00000041)* was upregulated in two RNA-seq datasets of rice under heat stress, with 99.8% similarity to the *ORUFI11G23900.1* gene in *Oryza rufipogon*. The cysteine-rich receptor-like protein kinase 6 (*maker_00001878*) gene upregulated in two RNA-seq datasets was homologous to the *OBART07G17510.1* gene in *Oryza barthii*, with an identity of 81.4%. The thiol methyltransferase 2 domain-containing protein (*maker_00001393)* gene was homologous to the *OGLUM03G40460.1* gene in *Oryza glumipatula*. The sulfotransferase *(maker_00000647)* gene was identified to be homologous to *ONIVA11G17360.1* in *Oryza nivara*, with a similarity index of 100.0% (Figure 6 and Appendix A). 

### 3.7. Mapping DEGs to the Known Heat Stress-Tolerant QTLs

For further validation, we mapped the above DEGs with heat stress tolerance-related SNPs to the known heat stress-tolerant quantitative trait loci (QTLs) in rice. The positions of 63 heat stress-tolerant QTLs were retrieved from previous research. After filtering the overlapping QTL regions, 46 heat stress-tolerant QTL regions were selected (Appendix A). In order to identify the heat stress-tolerant candidate genes in the non-reference contigs, we mapped the heat stress-tolerant QTLs to the 12 different representative genomes of Asian domesticated rice cultivars [56]. The number of genes in the non-reference contigs mapped to the QTL regions in the different rice reference genomes varied between 37 and 60 genes (Table 4). Subsequently, we mapped the DEGs with heat stress tolerance-related SNPs to the heat stress resistance QTL regions. A total of 1677 DEGs, including 990 upregulated and 687 downregulated genes, were mapped to the 46 QTL regions, in which 2 upregulated genes were identified in the non-reference contigs. One of the genes was annotated as the protein transport protein Sec24-like and the other was root phototropism protein 2-like. Homology search of the Sec24-like protein showed that this gene is similar to the *ONIVA11G05100.1* gene from the wild species *Oryza nivara*. 

## 4. Discussion

The sequencing and assembly of the rice genome have allowed tremendous progress in rice genotyping and gene identification. Multiple studies have been conducted on rice using the pan-genome approach to mine the overall variation in rice cultivars, such as Zhao et al. (2018) [38], Wang et al. (2018) [66], Sun et al. (2016) [67], etc. These studies systemically investigated the whole set of coding genes in the pan-genome, which showed an extensive presence and absence of variation among the different rice varieties. On the other hand, the previous research identifying heat resistance-related variations in rice was based on a single reference genome, which might lose the genome structural variation information, including the presence/absence or copy number variation among the different individuals. The published rice reference genome assembly is 373 Mbp in size with 37,860 predicted genes [68]. In this study, we constructed a rice pan-genome from heat-responsive cultivars to identify and characterize the heat-tolerant candidate genes, especially those that are not present in the single rice cultivar reference genome. The pan-genome represents the entire gene set of heat stress-resistant and -susceptible rice cultivars, including core and variable genes. Compared to the single reference genome, the pan-genome constructed in this research had an increment in the genome size of 15.8% and an additional 1141 non-reference genes. This increment in additional genes was mainly due to the non-reference contigs, which could not be successfully mapped to the single reference genome, and these genes were annotated using additional EST and RNA-seq data evidence. The iterative mapping and assembly approach has been used to build pangenome references in facilititating the characterization of resistant genes in *Brassica napus* [69], and identification of number of agronomic trait-related genes in pigeon pea [43] and sorghum [42]. Here, it was used to construct the rice pan-genome, followed by remapping of the sequencing data to the pan-genome to identify the presence/absence variations in heat stress responsive varieties. 

Overall, 20.79% of the rice pan genomic genes were variable genes, and the PAV-based classification of the tested rice varieties was in agreement with the SNP-based cluster. A total of 159 and 370 unique genes were found in the resistant varieties in the two PAV-based clusters, respectively (Figure 2), demonstrating the structural variation among the heat stress-responsive rice cultivars. This was consistent with the multiple previous studies. Gabur et al. (2020) found the association of the gene PAV with *Verticillium longisporum* disease resistance in *Brassica napus* [70]. In another study, Weisweiler et al. (2019) applied the transcriptomic data and PAV in the barley genome to predict phenotypic traits [71]. Therefore, PAV in the pan-genome might contribute to the phenotypic diversity of the heat stress-responsive rice cultivars.

With the rapid development of next-generation sequencing technologies, it is now much more reliable to discover DNA polymorphisms at a genome-wide scale, which plays a vital role in unraveling the genetic basis of phenotypic differences. As a result, the variant analysis using the pan-genome reference enabled us to discover 191,187 additional SNPs from the genetically diverse rice accessions. Recent pan-genomic studies, such as the studies by Ruperao et al. (2021) on the sorghum pan-genome [42], Zhao et al. (2020) on the pigeon pea pan-genome [72], and Li et al. (2021) on the cotton pan-genome, identified significantly associated SNPs in the non-reference sequences of the pan-genome. Thus, the SNPs on the non-reference contigs identified in this study are an added resource for identifying additional markers of heat tolerance in rice. Furthermore, the SNPs in the tested rice varieties’ genome were grouped into two clusters, which was consistent with the rice sub-groups, *Indica* and *Japonica*. Previous studies, such as Xu et al. (2020), found significant variation in the *LOC_Os12g39840* (SLG1) gene between *Japonica* and *Indica* species, which confer high-temperature tolerance in *Indica* rice [72]. Given the fact that the rice plants’ resistance to heat stress varies with their genetic background [73], the heat-tolerant SNPs detected in this study are of great value for further genotype–phenotype studies and useful for the breeding of new heat-tolerant rice varieties. 

Genome-wide SNP markers have been used to identify stress-resistant genes in plants in previous studies, including Li et al. (2017), Silva et al. (2012), and Xu et al. (2014), etc. [74,75,76]. These studies identified a number of stress-resistant candidate genes and SNPs using whole-genome and transcriptome comparison methods. In this study, using the constructed rice pan-genome as a reference, we identified 24 DEGs with heat tolerance-related SNPs in the non-reference contigs, including 15 upregulated and 9 downregulated genes (Figure 6). The functional annotation revealed that these genes might play a key role in heat stress tolerance. Among the upregulated genes, calmodulin-binding protein 60 A-like (CAM), a ubiquitous and multifunctional Ca^2+^ sensor, was involved in heat stress tolerance in a number of plant species, including *Arabidopsis* [77,78,79]. PDR-like ABC transporter is known to play a key role in cellular signaling and environmental adaptation. Rizhsky et al. (2004) reported that the expression of ABC transporter in *Arabidopsis* was enhanced by multiple stresses, especially heat and drought [80]. Sulfotransferases (SOTs) are sulfate-regulating proteins found in various organisms. Chen et al. (2012) analyzed the genome-wide comprehensive expression of 35 putative SOT genes in rice and characterized 11 SOTs that participated in the response to abiotic stresses [81]. Other genes among the 24 DEGs with heat tolerance-related SNPs in the non-reference contigs, including proteasome subunit α type-5 gene, protein transport Sec24, cell division cycle gene, senescence-related genes (SRGs), cysteine-rich receptor-like protein kinase 6 (CRK6), and cytochrome P450 genes, were also found to be involved in the response to stresses [82,83,84,85,86]. The other 15 genes might be novel heat stress resistance genes in rice and will be confirmed in our functional validation experiments. Therefore, the combination of SNP detection and transcriptome analysis was an effective approach to discover the novel heat stress-tolerant candidate genes in rice.

Several heat-resistant QTLs in rice have been identified in previous research (Appendix A). In this study, we also mapped the rice pan-genome genes to the heat-resistant QTLs in different rice reference genomes to identify the heat stress resistance candidate genes. We used the strategy of combining the SNP detection and RNA-seq data analysis with previously identified QTL regions, which were broadly applied to identify the key candidate genes corresponding to the different traits in different crops. Wen et al. (2019) used a similar strategy to identify the stress-resistant genes in tomato while Behnam et al. (2020) applied this method to detect the candidate genes associated with cadmium tolerance in barley [87,88]. Additionally, previous pan-genome studies also found a number of non-reference genes corresponding to different agronomic traits. For example, Ruperao et al. (2021) identified 79 genes associated with drought stress in sorghum [42], Li et al. (2021) uncovered 124 PAVs linked to a favorable fiber quality and yield loci [89], etc. In this study, among the 24 non-reference DEGs with heat resistance-related SNPs, we identified 2 upregulated genes, which were annotated as protein transport protein Sec24-like and root phototropism protein 2-like, that were mapped to known heat stress-tolerant QTL regions. Protein transport Sec24 are components of the COP II complex response during the ER-to-Golgi transport of secretory proteins. Qian et al. (2015) found that multiple genes of this family were upregulated in response to different abiotic stress treatments in rice [90]. In conclusion, the findings of this study provide insight into the further functional characterization of the heat resistance candidate genes identified in the non-reference contigs in rice.

## 5. Conclusions

We constructed and characterized the rice pan-genome using the rice reference genome and the whole-genome resequencing reads of 60 heat stress-responsive rice varieties. The pan-genome had 38,898 genes, which were categorized into core and variable genes according to the presence and absence variation. The results showed that PAV in the pan-genome contributed to the phenotypic diversity of the heat stress-responsive rice cultivars. Consequently, 3214 upregulated and 2212 downregulated genes with heat tolerance-related SNPs were identified by combining the strategy of SNP and transcriptomic analysis. Twenty-four DEGs with heat resistance-related SNPs were located in the non-reference contigs of the pan-genome, among which most were annotated as stress-responsive genes in rice. Two DEGs with heat resistance-related SNPs in the non-reference contigs were mapped to the known heat-resistant QTLs. Overall, the results of this study provide further insight for researchers on the functional validation of these heat stress resistance candidate genes.

## Figures and Tables

**Figure 1 genes-13-01353-f001:**
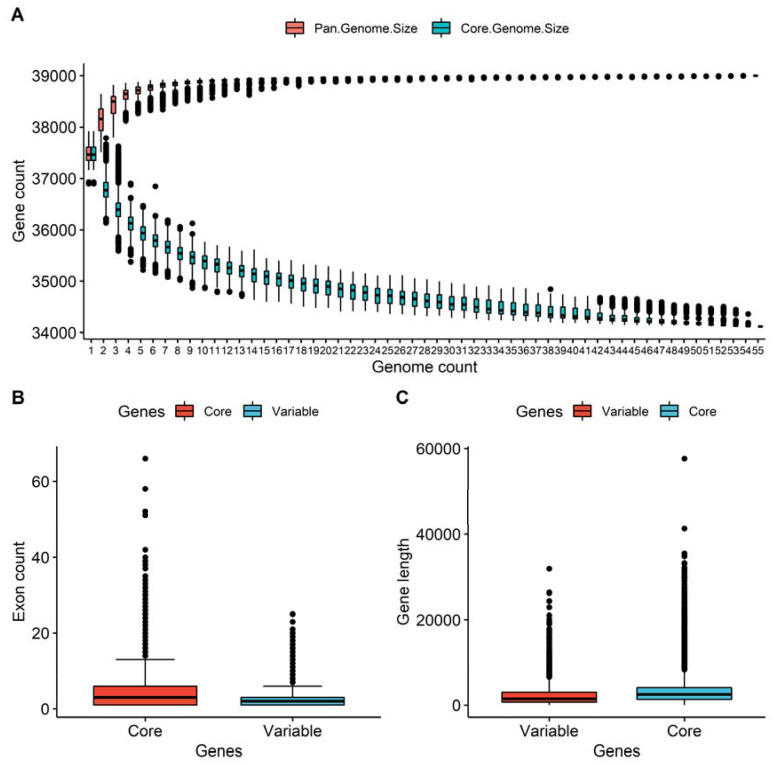
Gene distribution in the rice pan-genome. (**A**) The growth model of pan-genome genes and core genes. (**B**) Distribution of exon counts among the core and variable genes. (**C**) Distribution of the length of genes among the core and variable genes.

**Figure 2 genes-13-01353-f002:**
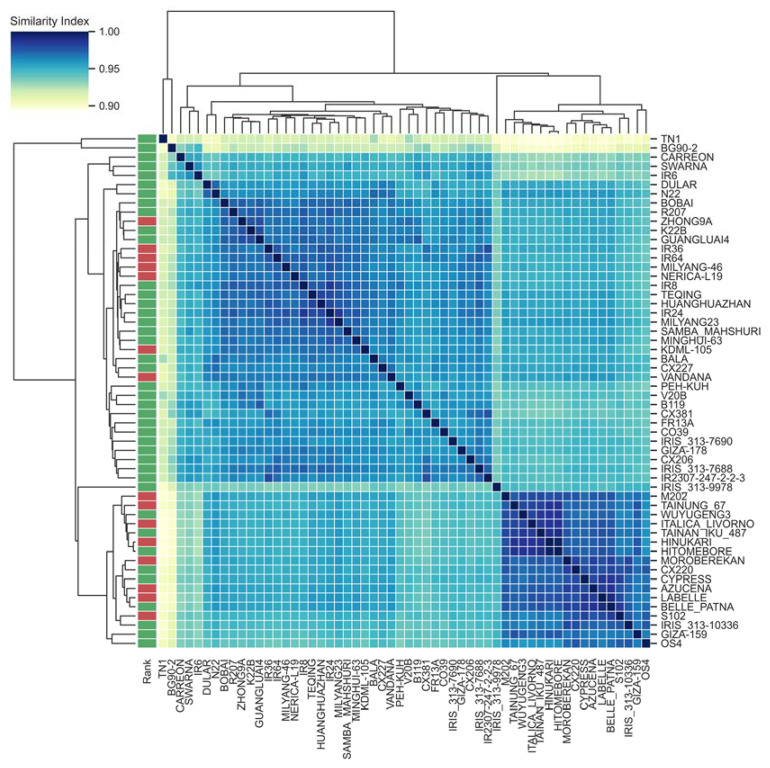
PAV-based pairwise relationships among the tested rice accessions *. * The left column indicates the heat stress response of the different rice cultivars. Green and red represent the heat-tolerant and -susceptible cultivars, respectively.

**Figure 3 genes-13-01353-f003:**
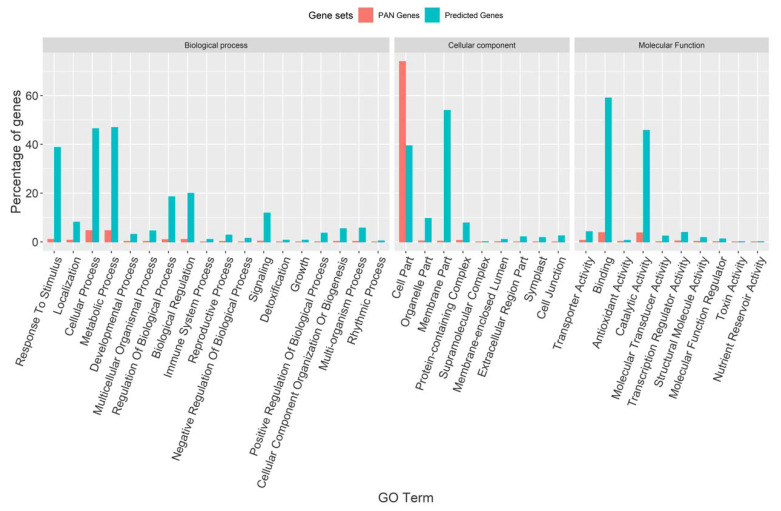
Functional annotation of the predicted genes in the non-reference contigs of the rice pan-genome. GO terms enriched at padj < 0.05 significance.

**Figure 4 genes-13-01353-f004:**
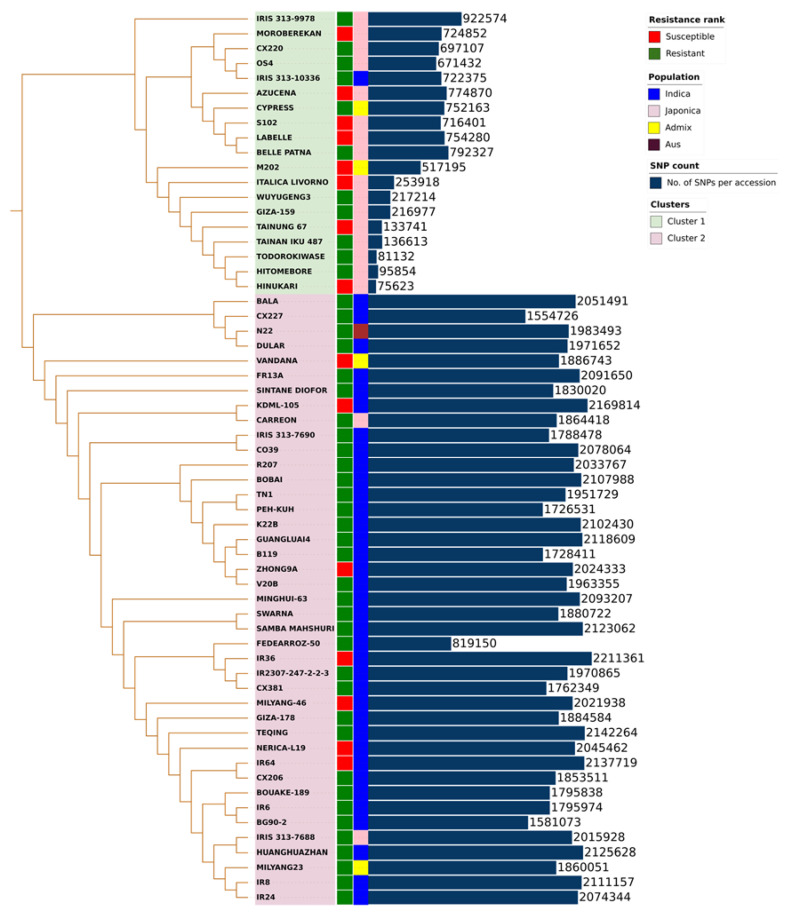
SNP-based phylogenetic tree of the tested rice varieties. The inner strip represents the heat stress response (R for resistant and S for susceptible) while the outer one represents the population of each rice cultivar based on their metadata. The outer bar chart represents the number of SNPs in each rice accession.

**Figure 5 genes-13-01353-f005:**
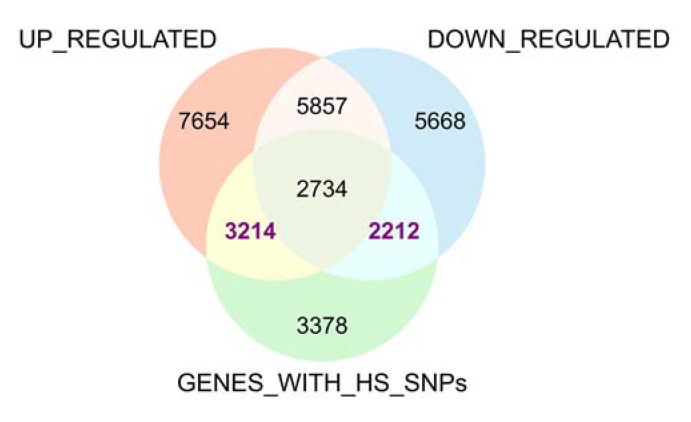
**A** Venn diagram plot of the genes with heat tolerance-related SNPs (GENES_WITH_HS_SNPs) compared to the upregulated and downregulated genes in the 4 RNA-seq datasets.

**Figure 6 genes-13-01353-f006:**
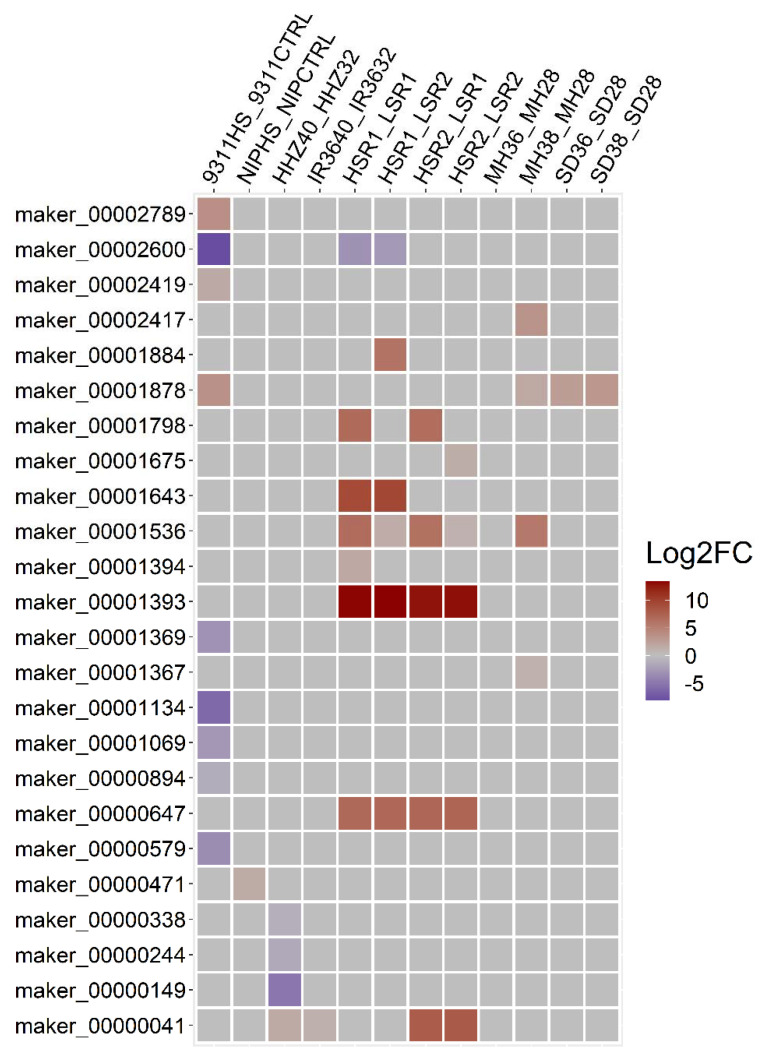
Differential expression profile of the genes with heat stress tolerance-related SNPs in different transcriptomic datasets of rice under heat stress.

**Table 1 genes-13-01353-t001:** Classification of the SNPs detected in the rice pan-genome.

Variant Type	Pan-Genome	Reference	Non-Reference	Cluster 2	Cluster 1
Bi-allele SNP	5,059,798	4,868,611	191,187	4,763,997	2,373,085
Splicing	15,986	15,881	105	15,210	7773
Exonic	284,016	281,185	2831	269,738	140,517
Intronic	578,887	576,534	2353	548,909	277,418
UTR	224,599	224,127	472	213,375	105,933
Upstream	1,229,370	1,224,403	4967	2,790,074	1,351,465
Downstream	1,100,851	1,095,937	4914	2,595,947	1,262,776
Missense	143,819	142,071	1748	136,135	70,833
Stop gained	1936	1900	36	1816	840

**Table 2 genes-13-01353-t002:** Distribution of the heat stress-tolerant SNPs in the rice pan-genome.

Annotation	SNPs in Resistant Cultivars	SNPs in Highly Resistant Cultivars
Downstream	29,082	14,759
Exon	358	185
Intron	18,846	9541
Non_Synonymous	10,194	5046
Splice site acceptor	28	16
Splice Site donor	38	19
Start gained	769	366
Start lost	25	11
Stop gained	225	116
Stop lost	26	14
Synonymous	7827	4090
Upstream	66,718	35,970
UTR_3	8820	4422
UTR_5	3817	1880
**Total**	**146,773**	**76,435**

**Table 3 genes-13-01353-t003:** Numbers of DEGs in the 4 RNA-seq datasets of rice under heat stress.

ProjectID *	Test Cultivars	Comparisons	Pan-Genome Upregulated Genes	Reference Upregulated Genes	Non-Reference Upregulated Genes	Pan-Genome Downregulated Genes	Reference Upregulated Genes	Non-Reference Downregulated Genes
PRJNA604026	9311	9311HS_9311CTRL	8248	8202	46	9691	9616	75
	Nipponbare	NIPHS_NIPCTRL	4914	4909	5	6504	6495	9
PRJNA508820	Huanghuazhan	HHZ40_HHZ32	2091	2064	27	1819	1800	19
	IR36	IR3640_IR3632	1395	1364	31	1503	1486	17
PRJNA610667	HSR1	HSR1_LSR1	2143	2058	85	1650	1605	45
	HSR2	HSR1_LSR2	1704	1645	59	1048	1020	28
	LSR1	HSR2_LSR1	1617	1560	57	2232	2180	52
	LSR2	HSR2_LSR2	1596	1532	64	2117	2060	57
PRJNA633211	MH101	MH36_MH28	1825	1809	16	1361	1354	7
		MH38_MH28	5181	5145	36	2898	2881	17
	SDW005	SD36_SD28	1380	1375	5	110	103	7
		SD38_SD28	4618	4599	19	923	916	7

* NCBI project accession ID.

**Table 4 genes-13-01353-t004:** Number of genes in the non-reference contigs mapped to the heat-tolerant QTLs in each reference genome.

BioSample ID *	Number of Genes
SAMN08217222	37
SAMN10564385	60
SAMN12715984	49
SAMN12721963	46
SAMN12672924	55
SAMN12718029	49
SAMN12748569	38
SAMN12748589	42
SAMN12748590	55
SAMN12748600	41
SAMN12748601	39
SAMN13021815	51

* Biosample ID of the rice accession.

## Data Availability

All data generated during this study are included in this published article and its Appendix A files.

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
