# Peer review of "Identification of Heat-Tolerant Genes in Non-Reference Sequences in Rice by Integrating Pan-Genome, Transcriptomics, and QTLs"

_genes, 2022, doi:10.3390/genes13081353_

Round 1
Reviewer 1 Report
The authors showed great work. The pan genome resource is an important step towards the effective and efficient genetic improvement of heat stress resistance in rice to help meet the rapidly growing needs for improving the rice roductivity under different environmental stresses. In this study, Pan-genomes has been constructed to provide insight into genome diversity and functionality of different plants which can be used in genome assisted crop improvement. Thus a pan-genome comprising of all genetic elements is crucial for comprehensive variation study among the heat resistant and susceptible rice varieties. A rice pan-genome was firstly constructed by using 45 heat-tolerant and 15 heat-sensitive rice varieties. A total of 38,998 pan-genome genes were identified, including 37,859 genes in the reference and 1,141 in non-reference contigs. Genomic variation analysis demonstrated that a total of 76,435 SNPs were detected and identified as the heat tolerance related SNPs, which specifically present in the highly heat resistant rice cultivars and located in the genic regions or within 2kbp upstream and downstream of the genes.
And also, 3,214 up-regulated and 2,212 down-regulated genes with heat stress tolerance related SNPs were detected in one or multiple RNA-seq datasets of rice under heat stress, among which 24 were located in the non-reference contigs of the rice pan-genome. And they mapped the DEGs with heat stress tolerance related SNPs to the heat stress resistant QTL regions. A total of 1,677 DEGs, including 990 up and 687 down-regulated genes, were mapped to the 46 heat stress resistant QTL regions, in which 2 up-regulated genes with heat stress tolerance related SNPs were identified in the non-reference sequences. These findings provide further insight for functional validation of number of non-reference genes and especially the 2 genes identified in the heat stress resistance QTLs in rice.
Author Response
Dear Reviewer,
Thanks a lot, we really appreciate your valuable comment
Reviewer 2 Report
Comments:
1. Page 4, lines 195-196: Authors divided these 60 rice accessions with different tolerant abilities to heat stress into five groups: highly resistant, resistant, moderately resistant, susceptible, and highly susceptible. Please provide more details about the criteria for the classification. And what is the meaning of this classification?
2. Page 5, lines 219-220: “rice cultivars showed that 54 genes were uniquely present in the heat resistant rice cultivars, including 5 genes from the reference contigs whereas 48 were from the additional non-reference contigs”, there are only 53 (5+48) genes, but they addressed those 54 genes were uniquely present?
3. Page 6, Figure 1 legend was different with figure content: should correct to (B) Distribution of exon counts among the core and variable genes. (C) Distribution of the length of genes among the core and variable genes.
4. Page 7, lines 251-255: Authors described the results of functional annotation of genes in different ways? Such as 38.9%, 205, 11.8%, 535, and so on. Please unify these descriptions, we can’t point out the details from Figure 3.
5. Page 8, lines 279-281: “The first cluster contained 41 rice varieties in which the majority belonged to Indica species, whereas the second cluster included the remaining 19 rice varieties with majority belonging to Japonica species”. The descriptions are different from Figure 4? The majority of the first cluster belonged to the Japonica species and the majority of the second cluster belonged to the Indica species.
6. Please provide why they put IRIS 313-10336(Indica species) into the first cluster? Similarly, why are they putting CARREON and IRIS 313-7688 into the second cluster?
7. Page 12, line 351: Authors addressed “teasome subunit alpha type-5 (maker_00001068) were respectively homologous”, we can’t find (maker_00001068) in both Figure 6 & Supplementary materials Table S6?
8. There are several genes with heat stress-tolerant related SNPs in figure 6, such as marker_00001798, marker_00001643, and marker_00001536. Why do the authors not point them out for further analysis?
9. Please explain the difference between the (maker_00001068) putative STF-1 list in Table S6 and the (maker_00001068) sulfotransferase list in line 350. There are many family members in sulfotransferase.
Author Response
Dear reviewer,
First of all, we would like to thank you for you valuable comments.
We are responding to your comments as follows:
- Page 4, lines 195-196: Authors divided these 60 rice accessions with different tolerant abilities to heat stress into five groups: highly resistant, resistant, moderately resistant, susceptible, and highly susceptible. Please provide more details about the criteria for the classification. And what is the meaning of this classification?
We included the classification criteria in data collection (Materials and Methods)
- Page 5, lines 219-220: “rice cultivars showed that 54 genes were uniquely present in the heat resistant rice cultivars, including 5 genes from the reference contigs whereas 48 were from the additional non-reference contigs”, there are only 53 (5+48) genes, but they addressed those 54 genes were uniquely present?
We apologize for the mistake we made; it is corrected to 53 uniquely present in resistant varieties.
- Page 6, Figure 1 legend was different with figure content: should correct to (B) Distribution of exon counts among the core and variable genes. (C) Distribution of the length of genes among the core and variable genes.
We apologize for the mistake we made, it is corrected now.
- Page 7, lines 251-255: Authors described the results of functional annotation of genes in different ways? Such as 38.9%, 205, 11.8%, 535, and so on. Please unify these descriptions, we can’t point out the details from Figure 3.
These paragraph is completely modified to reflect the grouping and the numbers in each category.
- Page 8, lines 279-281: “The first cluster contained 41 rice varieties in which the majority belonged to Indica species, whereas the second cluster included the remaining 19 rice varieties with majority belonging to Japonica species”. The descriptions are different from Figure 4? The majority of the first cluster belonged to the Japonica species and the majority of the second cluster belonged to the Indica species.
We apologize for the mistake we made, it is corrected now.
- Please provide why they put IRIS 313-10336(Indica species) into the first cluster? Similarly, why are they putting CARREON and IRIS 313-7688 into the second cluster?
Although the meta-data of these cultivars showed that these two cultivars belong to Indica species, but the SNP-based classification showed that they belong to different cluster. Similar result was obtained in another study conducted on 3000 rice genomes data (https://cgm.sjtu.edu.cn/3kricedb/rice-table.php?action=submit).
- Page 12, line 351: Authors addressed “teasome subunit alpha type-5 (maker_00001068) were respectively homologous”, we can’t find (maker_00001068) in both Figure 6 & Supplementary materials Table S6?
We apologize for the mistake we made, although this gene was among the differentially expressed genes but since it does not harbor heat stress resistance related SNPs it was excluded from the final list of DEG. Therefore, we would like you to disregard this gene. We corrected the paragraph to reflect this change.
- There are several genes with heat stress-tolerant related SNPs in figure 6, such as marker_00001798, marker_00001643, and marker_00001536. Why do the authors not point them out for further analysis?
Although they are differentially regulated, they are associated with disease resistance. We gave more emphasis to the other abiotic stress and heat stress related genes as a result we didn’t point them to be considered for further analysis.
- Please explain the difference between the (maker_00001068) putative STF-1 list in Table S6 and the (maker_00001068) sulfotransferase list in line 350. There are many family members in sulfotransferase.
Although there are a number of STF (sulfotransferase) family genes in Table S6, there is only one gene (maker_00000647), which was among the 24 differentially expressed genes. As it is mentioned in Table S6, maker_00001068 is a proteasome subunit alpha type-5 gene family. We also responded to comment 7 that, we would like to disregard this gene from line 350.

Round 2
Reviewer 2 Report
I agree to accept the version 2.